# The Performance of Lateral Flow Tests in the Age of the Omicron: A Rapid Systematic Review

**DOI:** 10.3390/life12111941

**Published:** 2022-11-21

**Authors:** Qin Xiang Ng, Yu Liang Lim, Ming Xuan Han, Seth En Teoh, Julian Thumboo, Ban Hock Tan

**Affiliations:** 1MOH Holdings Pte Ltd., 1 Maritime Square, Singapore 099253, Singapore; 2Health Services Research Unit, Singapore General Hospital, Singapore 169608, Singapore; 3Department of Community Emergency Health and Paramedic Practice, Monash University, Clayton, VIC 3800, Australia; 4Yong Loo Lin School of Medicine, National University of Singapore, 10 Medical Dr., Singapore 117597, Singapore; 5Department of Rheumatology and Immunology, Singapore General Hospital, Singapore 169608, Singapore; 6SingHealth Duke-NUS Medicine Academic Clinical Programme, Duke-NUS Medical School, Singapore 169857, Singapore; 7Department of Infectious Diseases, Singapore General Hospital, Singapore 169608, Singapore

**Keywords:** diagnostics, antigen rapid test, lateral flow test, immunoassay, COVID-19, coronavirus

## Abstract

Prompt detection and isolation of COVID-19 cases is vital for preventing further viral transmission, and lateral flow or rapid antigen tests have been an important diagnostic tool in this pandemic. However, concerns have emerged regarding the sensitivity of these devices for the new BA.1, BA.2, and BA.4/5 omicron variants, which have greater transmissibility and extensive mutations in its spike (S) and nucleocapsid (N) proteins. N protein is an important target protein for existing lateral flow devices. This paper therefore aimed to provide a rapid review of available literature on the performance of the lateral flow tests for detecting the omicron coronavirus variant. A systematic literature search of PubMed, EMBASE, OVID Medline, and Google Scholar found six published studies and four preprints investigating the performance of existing lateral flow devices for the omicron variant, as compared to the B.1.617.2 (Delta) variant. Overall, it appears that the devices have poorer performance for the omicron variant and when testing samples with cycle threshold (Ct) values greater than 25 and in asymptomatic individuals. As most available data were preliminary and had small sample sizes, it is recommended that these data be further studied to better inform and adapt our public health responses.

## 1. Introduction

Today, many countries are seeing a surge in Coronavirus Disease 2019 (COVID-19) cases due to the emergence of the highly transmissible BA.1, BA.2, and BA.4/5 omicron coronavirus variants [1], which have overtaken the B.1.617.2 (delta) variant in frequency and currently dominates. In response to the highly transmissible omicron variant, several countries have employed lateral flow or rapid antigen tests with greater adoption, and emphasized more regular self-tests or point-of-care testing for the general populace [2]. Although user-friendly, convenient, and low-cost, the accuracy of these tests has become a matter of concern due to the large number of mutations in the spike (S) protein of the omicron variant, which is responsible for viral infectivity, immune evasion, and can affect viral load [3]. The omicron variant has more than 30 mutations compared to the earlier variants, and at least four mutations in nucleocapsid (N) protein have also been found [3], which is the primary target protein for almost all lateral flow tests. For reference, the delta variant has known D63G, R203M, and D377Y N protein mutations [4], while omicron has P13L, Δ31–33, R203K, and G204R N protein mutations [3], although the implications of these mutations for lateral flow or quick antigen tests remain unknown.

Lateral flow tests are immunoassays that work via the binding of conjugated antibodies to a specific antigen in a biological sample; in this case, direct antigen-based testing for the severe acute respiratory syndrome coronavirus 2 (SARS-CoV-2) N protein in a throat or nasal swab sample [5]. Briefly, these tests are immunochromatographic assays typically containing monoclonal antibodies to the N protein of the SARS-CoV-2 virus [5,6]. In a given sample, if SARS-CoV-2 is present, anti-SARS-CoV-2 monoclonal antibodies would bind to the viral antigens to form an antigen-antibody complex. This complex is then captured by anti-SARS-CoV-2 monoclonal antibodies immobilized on the “Test” line, and a line becomes visible on the test device [6]. The current gold standard for the detection of SARS-CoV-2 is a reverse transcriptase polymerase chain reaction (RT-PCR), however, this is hampered by the higher costs, greater expertise, and longer turnaround times required to run these samples [7]. In many countries, lateral flow test kits are thus widely distributed to households and the general population to enable rapid and more frequent self-testing [8], without the need for specialist providers or laboratory capacity. However, the sensitivities and specificities of lateral flow test kits have been known to be variable depending on the manufacturer or brand used [9], and the performance of these lateral flow devices have come under further scrutiny for their ability to detect the omicron coronavirus variant [10] due to the variant’s extensive mutations. Moreover, there are more than 200 different commercial brands of SARS-CoV-2 antigen test kits on the market today.

The performance of lateral flow devices in detecting the omicron variant has important policy and public health implications. This paper therefore aimed to provide a rapid review of available literature on the performance of the lateral flow tests for detecting the omicron coronavirus variant.

## 2. Methods

A systematic literature search was performed in accordance with the latest Preferred Reporting Items for Systematic Reviews and Meta-Analyses (PRISMA) guidelines [11]. By using the following combinations of broad Major Exploded Subject Headings (MeSH) terms or text words [lateral flow OR rapid antigen OR immunoassay] AND [covid OR coronavirus OR COVID-19 OR SARS-CoV-2], a comprehensive search of PubMed, EMBASE, OVID Medline, and Google Scholar databases yielded 11 papers published in English between 1 January 1988 and 1 June 2022. Attempts were made to search grey literature using the Google search engine. Titles and abstracts of records were downloaded and imported into EndNote bibliographic software and from there to the Covidence online tool (Vertitas Health Innovation Ltd., Melbourne, VIC, Australia) to streamline our systematic search process, and then screened by four independent researchers (Q.X.N., Y.L.L., M.X.H., S.E.T.). Full texts were obtained for all abstracts of relevance and their respective reference lists were hand-searched to identify additional relevant articles. Forward searching of prospective citations of the relevant full texts was also performed. Conflicts were resolved via consensus among the four researchers.

The inclusion criteria for this paper was any original study that investigated the laboratory or real-world performance of these devices for detection of the omicron variant versus the ancestral strains.

The relevant data were extracted using a standard data extraction form by three researchers (Y.L.L., M.X.H. and S.E.T.) and cross-checked by a fourth (Q.X.N.) for accuracy.

The quality of the studies reviewed were assessed using the Meta Analysis of Statistics Assessment and Review Instrument (MAStARI) appraisal tool [12] by consensus of three study researchers (M.X.H., S.E.T. and Q.X.N.).

## 3. Results

A total of 10 studies were reviewed. The search process was illustrated in Figure 1. Specific to the performance of lateral flow tests in detecting the omicron variant, six published report [10,13,14,15,16,17] and four preprint articles were found [18,19,20,21] (Table 1). Two were in vitro studies using viral isolates [10,15], one used Syrian hamsters inoculated with the variants [17], while the rest used anterior nasal or nasopharyngeal swab samples [13,14,16,18,19,20,21]. Most of the field studies came from the US [16,18,20,21], while one came from Belgium [13], one from France [14], and another from Switzerland [19].

All the devices reported worked based on qualitative detection of the N protein antigen from the SARS-CoV-2 virus. Most of these cross-sectional studies had a paucity of description of the study groups and a lack of consideration of potential confounding factors, e.g., analytical interferences and the antigenemia or viremia of certain individuals [22]. Further details on the risk of bias assessment can be found in Appendix A.

## 4. Discussion

Overall, there appears to be an appreciable decline in the performance of the lateral flow devices for the detection of the B.1.1.529 (omicron) coronavirus variant compared to the previous B.1.617.2 (delta) variant, especially during the first days following symptom onset. This is particularly evident with the older test kits, such as Standard Q and Panbio [19], although BinaxNOW^TM^ (which uses the same biologics and is only available in the US) showed comparable sensitivity for the omicron variant as previous variants based on two studies [15,20,21]. The LoD of these devices is likely lower in real-world antigen testing compared to laboratory validation studies. However, these data are still preliminary and four of the available papers were preprints that have not been formally peer-reviewed [18,19,20,21].

Based on an earlier modelling study of SARS-CoV-2, we know that very frequent testing (every 2 to 3 days) is required to have a meaningful impact on transmission dynamics [23]. Moreover, early epidemiological data suggest that the incubation period and serial interval of omicron are 4.2 days (range, 2–8 days) and 2.8 days (range, 1–7 days) respectively [24], which are shorter than the delta variant. Many countries have thus turned to the lateral flow or rapid antigen tests for cheap and quick testing, albeit there is still a lack of knowledge on viral load progression and infectivity over time especially for the new variants of concern (VOCs).

A 2021 systematic review on the test performance of lateral flow devices found that the reported sensitivities ranged from 37.7% (95% CI 30.6–45.5) to 99.2% (95% CI 95.5–99.9), and specificities from 92.4% (95% CI 87.5–95.5) to 100.0% (95% CI 99.7–100.0), with the performance of the devices being manufacturer-dependent rather than operator-dependent [9]. In general, the lateral flow devices have poorer performance for samples with cycle threshold (Ct) values greater than 25 and in asymptomatic individuals, most likely due to their lower viral loads [25]. The SARS-CoV-2 virus has previously been shown to have Ct values ranging from 18 to 40 in human oropharyngeal or nasal swab samples [26]. This is an important point to consider as many countries, such as the UK, advocate for the use of lateral flow tests even for asymptomatic individuals. The utility of these devices for asymptomatic infection is likely limited as the omicron variant currently dominates [27].

To overcome the reduced sensitivity of the lateral flow tests, it may be useful to recommend two consecutive days of negative tests or paired testing with a RT-PCR may be preferred, at least in individuals at higher risk of severe COVID-19 illness. Alternatively, one can also repeat the test with another brand device or at another setting if the first test yields a negative or inconclusive result. The associated costs must be balanced with the utility of a robust surveillance system for outbreak prevention, especially in high-risk settings, e.g., hospitals and intensive care units. Measures to clinch the diagnosis early may also be more valuable now that therapeutics are available, with the main benefit of anti-viral treatment being their ability to cut the rate of progression to severe disease, as shown in the use of remdesivir in early COVID-19 illness [28] and the Paxlovid studies [29]. Nonetheless, most of the lateral flow test kits still have acceptable test performance, especially for general widespread use, as they meet the sensitivity requirement of ≥80% and specificity requirement of ≥97% set by most regulatory agencies and the World Health Organisation (WHO) [30].

The findings of this rapid review should be interpreted in light of several limitations. There are undoubtedly challenges with regard to investigating a novel coronavirus variant in real-time, and the findings of preprint articles have not been formally peer-reviewed and are subject to change. Many of the reports also have relatively small sample sizes, and it is difficult to perform sensitivity evaluation of lateral flow devices without robust paired RT-PCR testing. Further validation studies are necessary. The performance of these devices for the detection of the omicron subvariants BA.2 and BA.4/5 also remains unknown as the available studies only examined the BA.1. It is also difficult to conduct side-by-side clinical performance comparisons of these clinical devices for the different variants due to the current absolute dominance of omicron globally. Further post-marketing surveillance and field investigations are needed to confirm these preliminary findings and draw precise conclusions.

However, in view of the emerging data, we should perhaps encourage the use of combined throat/nasal swabs to improve the performance of COVID-19 lateral flow tests. The anterior nasal (AN) swab is less invasive for patients but may give lower sensitivity compared to a nasopharyngeal (NP) sample. Based on earlier studies done on SARS-CoV-2, NP or combined nasal/throat swab testing had a greater diagnostic yield [31,32]. While it may be possible to self-administer nasal/throat swabs [33], NP sampling is uncomfortable even with good technique, and may be hard for the general population to self-administer at home. The swab may be withdrawn prematurely before it reaches the nasopharynx and become saturated with mucus, inevitably producing false results [34]. In deploying these devices for public health protection, their user-friendliness must also be considered.

## 5. Conclusions

As countries around the world move towards a new normal and resume cross-border travel and business as usual, lateral flow devices have become an important rapid diagnostic tool in this COVID-19 pandemic. Individuals who are unwell can promptly test and isolate themselves to prevent further disease transmission, especially to vulnerable persons, who may add to the strain of health systems by contracting the virus. However, as these practical tools may have decreased sensitivities for the detection of the omicron variant, a negative result should not be taken to be a confirmatory result. It is vital that these emerging data be further studied, so as to better inform and adapt our public health policies and responses.

## Figures and Tables

**Figure 1 life-12-01941-f001:**
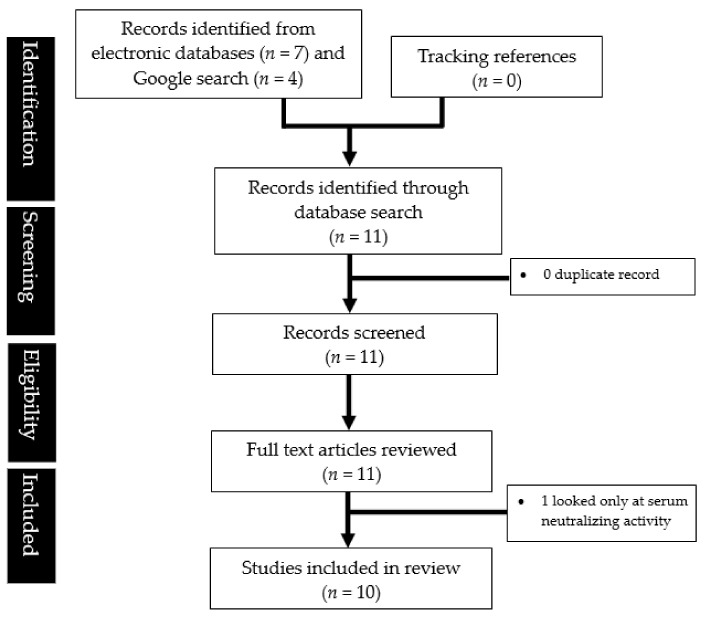
PRISMA flowchart showing the study selection process.

**Table 1 life-12-01941-t001:** Studies included in this review (arranged alphabetically by first author’s last name).

Study	Type of Lateral Flow Tests	Type of Swab Sample	Sample Size	Key Findings							
Adamson, 2022 [18] ^†^	Quidel QuickVue At-Home OTC COVID-19 Test Abbott BinaxNOW COVID-19 Antigen Self-Test	Anterior nasal	*n* = 30	Four cases were confirmed to have transmitted the virus between false-negative antigen tests On days 0 and 1, all rapid antigen tests produced false-negative results, despite 28 of 30 pairs having infectious viral load within the range of confirmed omicron transmissions in the cohort (Ct < 29) Swab samples were self-collected while observed by a trained COVID safety manager							
Bayart, 2022 [13]	Clinitest^®^ Rapid COVID-19 Antigen test New-Gene COVID-19 Antigen Detection Kit Boson Rapid SARS-CoV-2 Antigen Test Card Flowflex COVID-19 Antigen Home Test Sejoy SARS-CoV-2 Antigen Rapid Test Cassette Roche SARS-CoV-2 Rapid Antigen Test	Nasopharyngeal	*n* = 60	For samples with Ct ≤ 25, the calculated sensitivities were within the expected performance of these assays, largely similar among the devices and comparable between the omicron and delta variants. Apart from Roche, the other assays had significant variability and dismal performance for samples with Ct ≥ 25, especially for the omicron variant. *Reported test sensitivities (95% CI) for omicron and delta:*							
				**Population**	**Variant**	**Clinitest**	**New-Gene**	**Boson**	**Flowflex**	**Sejoy**	**Roche**
				Ct ≤ 25	Delta	95.6 (84.9–99.5)	95.6 (84.9–99.5)	97.8 (88.2–99.9)	97.8 (88.2–99.9)	95.6 (84.9–99.5)	100 (92.1–100)
					Omicron	94.1 (80.3–99.3)	97.1 (84.7–99.9)	97.1 (84.7–99.9)	91.2 (76.3–98.1)	97.1 (84.7–99.9)	100 (89.4–100)
				Ct ≥ 25	Delta	32.0 (15.0–53.5)	40.0 (21.1–61.3)	40.0 (21.1–61.3)	20.0 (6.8–40.7)	36.0 (18.0–57.5)	80.0 (59.3–93.2)
					Delta	32.0 (15.0–53.5)	40.0 (21.1–61.3)	40.0 (21.1–61.3)	20.0 (6.8–40.7)	36.0 (18.0–57.5)	80.0 (59.3–93.2)
Bekliz, 2022 [19] ^†^	Panbio COVID-19 Ag Rapid test device (Abbott) Standard Q COVID-19 Ag (SD Biosensor/Roche) Sure Status (Premier Medical Corporation) 2019-nCoV Antigen test (Wondfo) Beijing Tigsun Diagnostics Co. Ltd. (Tigsun) Onsite COVID-19 Ag Rapid Test (CTK Biotech) ACON biotech (Flowflex) NowCheck COVID-19 Ag test (Bionote)	Nasopharyngeal	*n* = 18	When assessing overall test positivity for clinical samples taken from vaccinated individuals infected with either omicron or delta, for omicron 124/252 (49.2%) of tests showed a positive result compared to 156/238 (65.5%) for the delta variant (z = −3.65, *p* < 0.001) Of 126 test pairs, 14 showed a discordant result for omicron vs. 7 in 119 test pairs performed for delta (z = −1.46, *p* = 0.144) When comparing sensitivity for delta vs. omicron for each Ag-RDT, four Ag-RDTs showed significantly lower sensitivity (*p* < 0.001) while three tests showed comparable performance Sensitivity in the specimens panel ranged from 22.2% to 88.9% for omicron and 52.9% to 91.2% for the delta variant *Reported test sensitivities (%) for omicron and delta:*							
				**Brand**		**Omicron**		**Delta**		***p* value**	
				Panbio		36.1%		67.6%		< 0.001	
				Standard Q		22.2%		52.9%		< 0.001	
				Sure Status		27.8%		52.9%		< 0.001	
				Onsite		47.2%		64.7%		< 0.001	
				Wondfo		75.0%		76.5%		0.984	
				Tigsun		47.2%		52.9%		0.634	
				Wondfo		75.0%		76.5%		0.984	
Deerain, 2021 [10]	Panbio™ COVID-19 Ag Rapid Test Device (Nasal) NowCheck COVID-19 Antigen Test SARS-CoV-2 Rapid Antigen Test STANDARD™ Q COVID-19 Ag Test Surescreen Diagnostics COVID-19 Antigen Rapid Test Cassette VivaDiag™ SARS-CoV-2 Ag Rapid Test Wantai SARS-CoV-2 Ag Rapid Test (Colloidal Gold) Testsea SARS-CoV-2 Antigen Test Kit InnoScreen COVID-19 Antigen Rapid Test Device LYHER Novel Coronavirus (COVID-19) Antigen Test Kit (Colloidal Gold)	In vitro study; cell cultures	Not applicable	This was an in vitro study. Overall, the analytical sensitivities were similar for both delta and omicron variants for the ten antigen test kits. All the antigen test kits were able to detect delta at Ct 25.4 and omicron at Ct 25.8, consistent with previous data.							
Gourgeon, 2022 [14]	COVID-VIRO antigen rapid test (AAZ-LMB) AMP rapid test SARS-CoV-2 Ag (AMP) Medakit Novel coronavirus (COVID-19) antigen test kit (Novel) BSD-0500333-25- COVID19 speed antigen test (Biospeedia) SARS-CoV-2 spike colloidal gold chromatographic assay (R-Biopharm) Test antigénique rapide Clinitest COVID-19 (Siemens) Abbott BinaxNOW™ COVID-19 Antigen Self-Test BIOSYNEX COVID-19 BSS test (Biosynex)	Nasopharyngeal	*n* = 179	Although there was a decline in test performance and greater likelihood of false negatives, especially during the first days following symptom onset (due to low viral loads), there was no significant difference in test sensitivity for all eight test kits when comparing omicron and the ancestral strain. *Reported test sensitivities for any Ct cutoff (95% CI):*							
				**Brand**		**Omicron (BA.1)**		**Delta**		**Alpha**	
				AAZ-LMB		70.0 (55.4–82.1)		88.9 (77.4–95.8)		88.9 (77.4–95.8)	
				AMP		70.0 (55.4–82.1)		90.7 (79.7–96.9)		90.7 (79.7–96.9)	
				Novel		70.0 (55.4–82.1)		86.5 (74.2–94.4)		86.5 (74.2–94.4)	
				Biospeedia		70.0 (55.4–82.1)		88.5 (76.6–95.6)		88.5 (76.6–95.6)	
				R-Biopharm		58.0 (43.2–71.8)		86.0 (73.3–94.2)		86.0 (73.3–94.2)	
				Siemens		68.0 (53.3–80.5)		88.9 (77.4–95.8)		88.9 (77.4–95.8)	
				Abbott		56.0 (41.3–70.0)		77.4 (63.8–97.7)		77.4 (63.8–87.7)	
				Biosynex		58.0 (43.2–71.8)		87.0 (75.1–94.6)		87.0 (75.1–94.6)	
				*Reported test sensitivities for Ct ≤30 (95% CI):*							
				**Brand**		**Omicron (BA.1)**		**Delta**		**Alpha**	
				AAZ-LMB		89.5 (75.2–97.1)		81.5 (68.6–90.7)		92.3 (81.5–97.9	
				AMP		92.1 (78.6–98.3)		92.6 (82.1–97.9)		94.2 (84.1–98.8)	
				Novel		89.5 (75.2–97.1)		87.0 (75.1–94.6)		90.0 (78.2–96.7)	
				Biospeedia		89.5 (75.2–97.1)		88.9 (77.4–95.8)		92.0 (80.8–97.8)	
				R-Biopharm		73.7 (56.9–86.6)		77.8 (64.4–88.0)		87.5 (74.8–95.3)	
				Siemens		86.8 (71.9–95.6)		85.2 (72.9–93.4)		92.3 (81.5–97.9)	
				Abbott		71.1 (54.1–84.6)		75.9 (62.4–86.5)		80.4 (66.9–90.2)	
				Biosynex		73.7 (56.9–86.6)		83.3 (70.7–92.1)		90.4 (79.0–96.8)	
Kanjilal, 2022 [20] ^†^	Abbott BinaxNOW™ COVID-19 Antigen Self-Test	Anterior nasal	*n* = 32	The BinaxNOW^TM^ test was positive in 9 of 22 (41%) delta samples and in 8 of 24 (33%) omicron samples with Ct < 30 It was positive in 1 of 7 (12%) delta samples and in 0 of 8 (0%) omicron samples with Ct ≥ 30 There were no statistically significant differences between the variants with respect to test positivity for either Ct range							
Schrom, 2022 [21] ^†^	Abbott BinaxNOW™ COVID-19 Antigen Self-Test	Anterior nasal, cheek and oral tonsillar	*n* = 296 (98.5% of a random sample of 75 persons were found to have the omicron variant)	BinaxNOW^TM^ rapid antigen test had comparable sensitivity for the omicron variant as prior variants *Reported test sensitivities (%) and specificities (%):*							
				**Ct cutoff**		**Sensitivity (95% CI)**				**Specificity (95% CI)**	
				30		95.2% (91.0–97.8%)				96.5% (94.6–97.9%)	
				35		82.1% (76.6–86.8%)				99.4% (98.2–99.9%)	
				No cutoff		65.2% (59.5–70.6%)				99.3% (98.0–99.9%)	
Stanley, 2022 [15]	Abbott BinaxNOW^TM^ COVID-19 Antigen Self-Test CareStart COVID-19 antigen (Access Bio) GenBody COVID-19 Ag test LumiraDx SARS-CoV-2 Ag test	In vitro study; live-virus culture	Not applicable	Overall, the limits of detection (LoD) were acceptable for both delta and omicron variants, albeit Abbott, GenBody, and LumiraDx test kits have ~2 orders magnitude better LoD for delta than the omicron. *Reported LoD (PFU/mL) of antigen tests based on live-virus cultures:*							
				**Brand**		**Omicron**				**Delta**	
				Abbott		8.3 × 10^1^				1.0 × 10^4^	
				Access Bio		2.8 × 10^3^				3.5 × 10^3^	
				GenBody		2.5 × 10^2^				3.5 × 10^4^	
Tsao, 2022 [16]	Abbott BinaxNOW™ COVID-19 Antigen Self-Test	Not specified; self-administered	*n* = 723 (95.7% (44 out of 46) positive cases were found to have the omicron variant)	Based on the RT-PCR findings, the overall sensitivity and specificity of the Ag-RDT were 63.0% (95% CI: 51.9–74.1) and 99.8% (95% CI: 99.5–100) respectively. The performance was similar to previous variants, where the lateral flow test had high specificity but poor sensitivity, especially among asymptomatic individuals.							
Weishampel, 2022 [17]	OraSure InteliSwab™	Live virus cultures and oropharyngeal swabs from hamsters inoculated with SARS-CoV-2	Not applicable	Overall, comparable test performance for alpha and omicron variant (*p* = 0.0385).							

^†^ Pre-print article. Abbreviations: COVID-19, Coronavirus Disease 2019; SARS-CoV-2, Severe Acute Respiratory Syndrome Coronavirus 2; Ct, Cycle threshold; 95% CI, 95% Confidence Intervals; Ag-RDT, Antigen-Detecting Rapid Diagnostic Test; LoD, limits of detection; RT-PCR, reverse transcriptase-polymerase chain reaction.

## Data Availability

Not applicable.

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
