# Peer review of "The Performance of Lateral Flow Tests in the Age of the Omicron: A Rapid Systematic Review"

_life, 2022, doi:10.3390/life12111941_

Round 1

Reviewer 1 Report

The authors have generally been able to show that the lateral flow tests perform worse for detecting Omicron as compared to Delta. However, these tests have mostly been given EUA approval based on their testing of the ancestral strain from Wuhan, from which both Delta and Omicron have mutated. Can the authors suggest why lateral flow tests have worse performance for detecting Omicron than Delta when both have mutated from the ancestral strain?

Line 15: The authors mentioned that lateral flow tests have comparable sensitivity and specificity to conventional real-time reverse transcription polymerase chain reaction (rt-PCR) tests. Don't think lateral flow is comparable to rt-PCR, as rt-PCR can detect 1-10 copies of SARS-CoV-2. Need to at least qualify which Ct range, e.g. < 25. 

Line 51:

- Is at least 2 mutations really a cause for concern? Also, authors may want to link how these mutations affect the performance of lateral flow tests. E.g. mutation in the binding epitope(s) of the antibodies.

- Since lateral flow tests target the N protein, are mutations on the S protein expected to have any significant effect on test performance?

- It would be interesting to know how differently the N protein has mutated for omicron vs. delta or other variants.

Line 54: Besides conjugated antibodies on the conjugate pad, there are also capture antibodies immobilized on the test line of the nitrocellulose test strip. Authors may want to provide a more complete description of the working principles of lateral flow tests.

Table 1, Adamson, 2022: This study has no comparison with other variants or wildtype to show that omicron results in reduced test accuracy. Also, it is highly unlikely to have all false-negative, as Ct < 29 is not an especially low viral load. And some countries, e.g. Singapore, require at least sensitivity of 80% for Ct < 30. Could be this an issue of lateral flow tests not conducted properly?

Table 1, Bekliz, 2022: 

- The N=18 sample size number does not match the numbers stated in the Key Findings column. 

- I'm not sure if test positivity is a meaningful parameter as there could just be fewer positive cases in the omicron cohort, and might not necessarily mean the tests are less accurate for omicron.

- What does 156/238 refer to? Delta variant?

- Authors should stratify the data according to Ct values as the reduced sensitivity could be due to lower viral loads. Or at least mention the Ct ranges for both omicron and delta.

Table 1, Gourgeon, 2022: Again, authors should stratify based on or mention Ct values.

Table 1, Stanley, 2022: Abbott, GenBody, and LumiraDx have ~ 2 orders magnitude better LoD for omicron than delta, and I wouldn't say that these are "similar" LOD.

Line 162:

- NP sampling is much harder for laymen to conduct at homes, and may lead to false negative due to poor sampling.

- Can authors comment on reports, which suggest a combination of throat and nasal swab leads to higher sensitivity?

Line 174:

- Can authors comment on how public health policies and responses still have room for lateral flow tests given that vaccination rates are quite high, at least for developed countries.

- How about for countries that have less access to vaccines then?

Author Response

Point-by-point replies

  1. Thank you for the comment. The word 'comparable' was based on available studies; we agree with the reviewer that the performance is only comparable when the Ct value is <25. We have now provided the data stratified according to Ct values in our results, as far as possible.
  2. We agree with the reviewer and have tried to better flesh out the differences between the Omicron and Delta variants, "The Omicron has more than 30 mutations compared to the earlier variants, and at least four mutations in nucleocapsid (N) protein have also been found [3], which is the primary target protein for almost all lateral flow tests. For reference, the Delta variant has known D63G, R203M, and D377Y N protein mutations [4], while Omicron has P13L, Δ31–33, R203K, and G204R N protein mutations [3]".
  3. We agree with the reviewer and have removed the comments on S protein mutations.
  4. We agree with the reviewer and have provided more information on the mutational changes, "For reference, the Delta variant has known D63G, R203M, and D377Y N protein mutations [4], while Omicron has P13L, Δ31–33, R203K, and G204R N protein mutations [3]" in our manuscript.
  5. Thank you for the comment. We have provided a greater explanation on how lateral flow tests work, "these tests are immunochromatographic assays typically containing monoclonal antibodies to the N protein of the SARS-CoV-2 virus [5,6]. In a given sample, if SARS-CoV-2 is present, anti-SARS-CoV-2 monoclonal antibodies would bind to the viral antigens to form an antigen-antibody complex. This complex is then captured by anti-SARS-CoV-2 monoclonal antibodies immobilized on the Test line, and a line becomes visible on the test device [6]."
  6. With regard to the comment on Table 1 Adamson 2022, we are unable to confirm any exact reasons; there is no indication that the swabs were performed improperly, we have provided additional information in the table that, "Swab samples were self-collected while observed by a trained COVID safety manager".
  7. For information, the study by Bekliz et al. (2022), the sample size is 18 although they took multiple clinical samples from these individuals for testing, hence the numbers don't seem to tally. When assessing overall test positivity for clinical samples taken from vaccinated individuals infected with either Omicron or Delta, for Omicron 124/252 (49.2%) of tests showed a positive result compared to 156/238 (65.5%) for the Delta variant (z = -3.65, p<.001). Of 126 test pairs, 14 showed a discordant result for Omicron vs. 7 in 119 test pairs performed for Delta (z = -1.46, p=.144). We have further stratified the test results for the study as suggested.
  8. Table 1, Gourgeon, 2022: thank you for the comment, we have further stratified the test results for the study as suggested.
  9. Table 1, Stanley, 2022: we totally agree with the reviewer's comments and have modified the text to read, "Overall, the limits of detection (LoD) were acceptable for both Delta and Omicron variants, albeit Abbott, GenBody, and LumiraDx test kits have ~2 orders magnitude better LoD for Delta than the Omicron."
  10. We agree with the reviewer's comments and have provided further elaboration on NP sampling vs combined throat/oral sampling. "The anterior nasal (AN) swab is less invasive for patients but may give lower sensitivity compared to a nasopharyngeal (NP) sample. Based on earlier studies done on SARS-CoV-2, NP or combined nasal/throat swab testing had a greater diagnostic yield [31,32]. While it may be possible to self-administer nasal/throat swabs [33], NP sampling is uncomfortable even with good technique, and may be hard for laypersons to self-administer at home. The swab may be withdrawn prematurely before it reaches the nasopharynx and become saturated with mucus, inevitably producing false results [34]. In deploying these devices for public health protection, their user-friendliness must also be considered."
  11. Thank you for the comments. Apart from prompt isolation, we have now added that the lateral flow tests also enable early detection and treatment, "Measures to clinch the diagnosis early may also be more valuable now that therapeutics are available, with the main benefit of anti-viral treatment being their ability to cut the rate of progression to severe disease, as shown in the use of remdesivir in early COVID-19 illness [28] and the Paxlovid studies [29]. "

Reviewer 2 Report

The authors make a systematic review on the efficacy of antigen tests to detect the Omicrom variant of SARS-CoV-2. Although the topic is not new, it is interesting since antigen tests release a lot of pressure on hospitals, as they are point of care tests. However, the Omicrom variant presents mutations in the S and N proteins, which are the ones on which most of these tests are based and hence the reasonable doubt raised by the authors about the real usefulness of the commercial tests analysed.

As I say, although I understand the opportunity of this study, I think there are some aspects that should be improved in this article:

-          On the one hand, include in the title that it is a systematic review.

-          In the methods section, the authors should explain the exclusion criteria. As they go from 176 articles found, how did they reduce these articles down to the 10 included in the study.

-          In the results section, they should indicate the databases to which the reviewed articles belong.

-          In addition, in Table 1 the authors should indicate whether each of these commercial tests is based on protein S or N because it could be significant when it comes to seeing the sensitivity of the test. And discuss this in the discussion section.

Otherwise, I agree with the authors. It is an interesting study, but right now there is not enough data, or at least those reported here, are not enough to be able to draw precise conclusions and a larger study that includes a meta-analysis is needed.

Author Response

Point-by-point replies

  1. Thank you for the comment. We have now included in our study title that this was a systematic review.
  2. We apologised for this mistake. To clarify, only 11 studies were found through extensive database searches, not 176.
  3. We have indicated which are the pre-print articles. The rest can all be found in PubMed.
  4. Thank you for the comment. We have now added in the results section that, "All the devices reported worked based on qualitative detection of the N protein antigen from the SARS-CoV-2 virus."